# IaC-Eval: A Code Generation Benchmark for Cloud Infrastructure-as-Code Programs

**Patrick Tser Jern Kon, Jiachen Liu, Yiming Qiu, Weijun Fan, Ting He**
**Lei Lin, Haoran Zhang, Owen M. Park, George S. Elengikal, Yuxin Kang**
**Ang Chen, Mosharaf Chowdhury, Myungjin Lee[‡], Xinyu Wang**
University of Michigan   [‡]Cisco Research

## Abstract

Infrastructure-as-Code (IaC), an important component of cloud computing, allows the definition of cloud infrastructure in high-level programs. However, developing IaC programs is challenging, complicated by factors that include the burgeoning complexity of the cloud ecosystem (e.g., diversity of cloud services and workloads), and the relative scarcity of IaC-specific code examples and public repositories. While large language models (LLMs) have shown promise in general code generation and could potentially aid in IaC development, no benchmarks currently exist for evaluating their ability to generate IaC code. We present IaC-Eval, a first step in this research direction. IaC-Eval's dataset includes 458 human-curated scenarios covering a wide range of popular AWS services, at varying difficulty levels. Each scenario mainly comprises a natural language IaC problem description and an infrastructure intent specification. The former is fed as user input to the LLM, while the latter is a general notion used to verify if the generated IaC program conforms to the user's intent; by making explicit the problem's requirements that can encompass various cloud services, resources and internal infrastructure details. Our in-depth evaluation shows that contemporary LLMs perform poorly on IaC-Eval, with the top-performing model, GPT-4, obtaining a pass@1 accuracy of 19.36%. In contrast, it scores 86.6% on EvalPlus, a popular Python code generation benchmark, highlighting a need for advancements in this domain. We open-source the IaC-Eval dataset and evaluation framework at https://github.com/autoiac-project/iac-eval to enable future research on LLM-based IaC code generation.

## 1   Introduction

Cloud computing has become a cornerstone of our digital infrastructure. According to recent reports [55, 31], 94% of all enterprises use cloud services of some form. Building on this trend, "Infrastructure as Code" (IaC) has emerged as the standard for developing cloud infrastructure. IaC allows users to codify their desired infrastructures within high-level IaC programs, which can be deployed repeatedly and consistently. The IaC frameworks in turn are responsible for provisioning the underlying resources (e.g., compute instances) specified in the program, by interacting with cloud-specific APIs. The most widely adopted tool leading this paradigm is Terraform [34], and is the focus of our paper. Terraform programs (otherwise known as configuration files) are written in the HCL language [68]; Fig. 1a illustrates a simplified example of such a program, where we construct an infrastructure consisting of a compute instance (VM) connected to a network. First, we have resource blocks which define instances of specific infrastructure components. Cloud services typically encompass multiple resources, each representing a distinct part of the service, and the selection of these resources depends on the specific use case; here, we need a `SUBNET` and `NIC`, both essential parts of the virtual private cloud service, a key networking component. Second, these resource blocks contain various attributes that dictate their instantiation. Attributes can have different

38th Conference on Neural Information Processing Systems (NeurIPS 2024) Track on Datasets and Benchmarks.

value types, including strings, enum types (e.g., "US-west"), lists (the value of `nic_ids`), and even nested structures (the `cpu_options` attribute). Third, these resources can be interconnected through specific attributes, forming a dependency graph. In our example, the three resources are interlinked: the `VM` is connected to the `NIC` via the `nic_ids` attribute, while the `NIC` and `SUBNET` are connected through the `subnet_id` attribute.

Developing IaC programs is a challenging task. For one, cloud infrastructures can be built using a vast array of services offered uniquely by various providers (e.g., AWS, GCP), each with complex domain-specific details/rules frequently underspecified at the IaC level [59], requiring deep cloud expertise to get right. Second, this could involve learning a new language, as is the case for Terraform, whose programs are specified in the feature-rich HCL language [68], which is a domain-specific language unfamiliar to most developers (e.g., as evidenced by the relative scarcity of IaC-specific code in public repositories compared to general-purpose languages [65]). Third, this complexity is further compounded by the growing diversity of cloud workloads incorporated within enterprises, extending beyond a few broad categories of Software-as-a-Service (SaaS) products [28], which in turn demands highly customized cloud infrastructures. Consequently, it is no surprise that 92% of enterprises employ extensive cloud engineering teams to manage this highly customized and intricate infrastructure [33].

We believe LLMs are a natural next step to aid in the process of creating infrastructure code. Indeed, LLMs have shown promise for general code generation, as exemplified by models such as Codex, CodeLlama [50], and AlphaCode [45]. As a result, a surge of datasets and benchmarks have emerged, such as the widely used hand-crafted HumanEval [26] (Python programming), to quantify how well these models perform. Unfortunately, while existing models and datasets/benchmarks target general-purpose languages, the ability of existing LLMs to effectively generate IaC code remains uncertain, since there exist no systematic studies nor datasets/benchmarks available to quantitatively evaluate LLM performance on IaC code generation.

**IaC-Eval** bridges this gap with the first dataset and benchmark for evaluating IaC code generation. As a first step in this domain, our dataset (Sec. 2.2) specifically targets AWS, the most popular cloud provider [67]. Our dataset includes 458 human-curated scenarios (comparable in size to the HumanEval dataset [26] which contains 164 programming problems) covering a variety of popular services, compiled over 1720 hours, ranging from simple to highly challenging scenarios that involve multiple resources across various services which can contain hundreds of lines of code (LoC). Each scenario consists mainly of a natural language problem description (e.g., "Create an AWS database") fed as user input to the LLM, and an infrastructure intent specification (Sec. 2.3) to check against the LLM generated program. This intent specification fulfils two key objectives:

*Objective 1: Determining user intent fulfilment.* The specification ensures that the generated IaC program conforms with the user's intent by explicitly detailing the problem's requirements, which can include various cloud services, resources, and internal infrastructure details. Crafting specifications is challenging, because just like regular programs: (1) User requirements for cloud infrastructure can be ambiguously specified. For instance, there are nearly a dozen ways to create an "AWS Database" [22] (e.g., DynamoDB, RDS). (2) Cloud infrastructure can also be complex. There are over 4,000 providers within the Terraform ecosystem alone [14], each offering unique services composed of resources utilized in various ways. For example, AWS alone provides more than 200 services, each with multiple configuration options and interdependencies. These services can sometimes involve dozens of interlinked resources, with specific instantiation details governed by provider-specific rules that may or may not be publicized [59]. Furthermore, with new services and updates continuously being introduced [23, 36], the landscape is constantly evolving, adding to the complexity. Overall, this means that the resulting specification for each problem description will codify a range of possible intent-fulfilling IaC programs, requiring deep cloud and IaC expertise to get right.

*Objective 2: Scalable evaluation.* Checking for intent fulfilment in IaC-Eval does not require executing the IaC programs, which typically involves deployment directly onto the cloud. Notwithstanding the fact that successful deployments are not an indication of correct fulfilment of user intention, this is problematic because deployments can take a significant amount of time—even simple configurations may require minutes to hours to deploy [59], making it an impractical evaluation strategy. Instead of waiting for a full deployment cycle (e.g., relying on knowledge from a state file [69] which will only be produced after a successful deployment), our evaluation benchmark relies only on compile-time operations that can be completed quickly and do not necessitate deployment (Sec. 2.1).

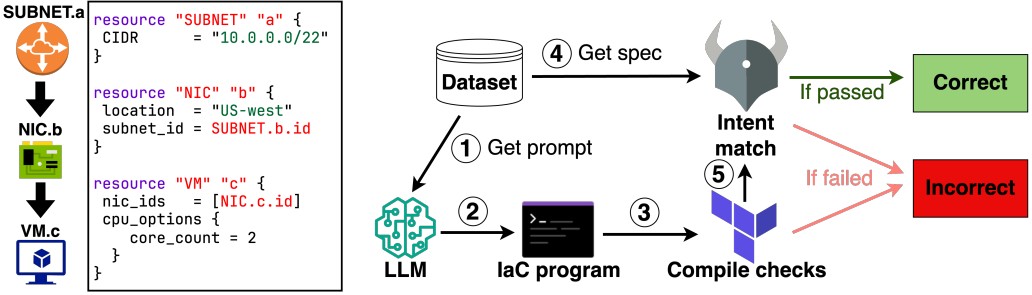

(a) Simplified example IaC program.    (b) IaC-Eval evaluation workflow.

Figure 1: IaC-Eval evaluation benchmark overview, and example IaC program.

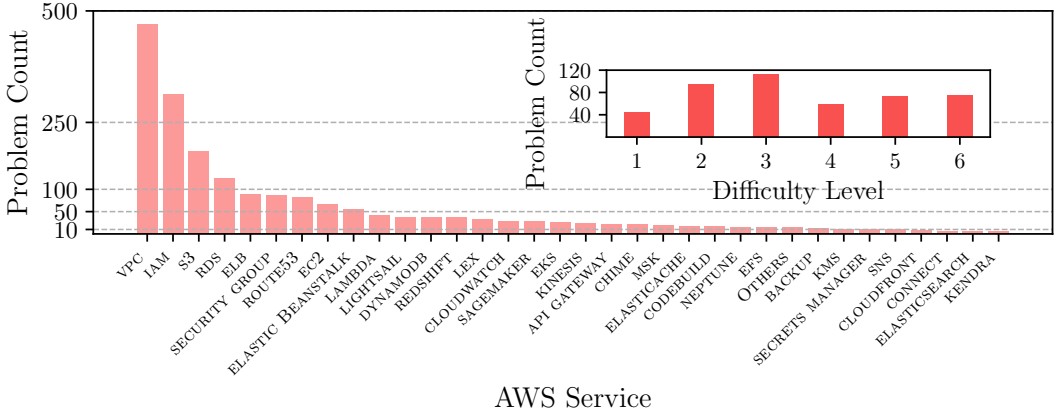

Figure 2: IaC-Eval dataset service and difficulty distribution.

**Comprehensive evaluation.** Finally, we conduct a detailed evaluation (Sec. 3) against a range of state-of-the-art models known for their strong performance on the EvalPlus [48] benchmark (an improved version of HumanEval). Our results show that these models perform poorly on IaC-Eval, even with various enhancement strategies.

## 2 The IaC-Eval benchmark and dataset

### 2.1 IaC-Eval benchmark: a high-level overview

We first describe our evaluation benchmark's overall workflow, illustrated in Fig.1b. Notably, we combined existing components natively supported by IaC (to benefit from continuous updates and robust support) into a novel benchmark for evaluating IaC code generation. ① First, a problem description is retrieved from the dataset (Sec. 2.2) and fed to the LLM under evaluation. ② The resulting LLM-generated IaC program is then fed into a two-phase pipeline that determines if the program is correct (or incorrect) without requiring deployment. ③ In the first phase, the generated IaC program is transformed into a speculative deployment plan using the native `terraform plan` [35] command, which produces a dependency graph. This phase includes basic validations to ensure syntactical accuracy and adherence to a limited set of cloud provider-specific requirements. Configurations failing this step are deemed incorrect by IaC-Eval. ④ Next, the problem's user infrastructure intent specification (Sec.2.3) written in the `Rego` language [51] is ⑤ matched against the dependency graph using `OPA` [66], the most widely used IaC policy engine; a tool typically used to define policies (e.g., security, conformance), that IaC-Eval repurposes to evaluate user intent fulfilment. If no errors are found, the IaC program is considered correct.

## 2.2 Dataset characteristics

IaC-Eval's dataset is structured with each row consisting of three key columns: (1) a natural language prompt describing the problem, (2) user intent specifications written in Rego (Sec. 2.3), and (3) one example of a correct configuration written in Terraform HCL [68], which we envision could be useful in the future for fine-tuning purposes. Creating this dataset requires significant domain knowledge and a deep understanding of cloud services and the intricacies of cloud infrastructure orchestration. It also demands proficiency in two languages markedly different from regular imperative languages: namely, HCL and Rego, which are complex declarative languages unfamiliar to regular developers.

Fig. 2 illustrates that our dataset addresses a comprehensive set of commonly used cloud infrastructure services [64, 56, 39], where each problem count is represented by some resource in a given service. This includes a full stack of infrastructure components typically required in cloud environments: (1) Compute services: EC2, Lambda, and Lightsail; (2) Relational and in-memory databases: RDS and ElastiCache; (3) Data warehouses: Redshift; (4) Networking elements: virtual private cloud (VPC), Route53 (DNS service), and API gateways; (5) Content delivery networks: CloudFront; (6) Key-value stores: DynamoDB; (7) Object storage: S3; (8) Security: identity and access management (IAM), and monitoring service (Cloudwatch); (9) Stream processing: Kinesis, and managed Kafka (MSK). We note that certain services (in particular, VPC and IAM) are often required across problems, and hence appear disproportionately in the dataset; in contrast, certain other services aren't composed of many distinct resources, and hence appear to have fewer counts in the dataset. Services are often interconnected in various configurations, requiring multiple services to address a single problem (e.g., an RDS instance deployed in a specific VPC). All of these services are also made up of a diverse set of lower-level resources that control more fine-grained functionality: for instance, as shown in Fig. 3(a), within the Aurora service, we could instantiate an Aurora DB cluster resource [20] (deployed across multiple regions), that utilizes a DB proxy [21] to pool/share database connections. Furthermore, each of these resources can be configured in unique ways through various default and optional attributes (e.g., setting an idle timeout duration for the DB proxy).

Finally, we introduce a system of difficulty levels for IaC problems. We recognize that determining these levels is inherently ambiguous and subjective, akin to the informal designations used by online programming platforms (e.g., LeetCode [18]) and in existing research [49]. Nevertheless, we propose an approximation that can calculate a difficulty level automatically by parsing the configuration, that is based on LoC, the number of resources, and their interconnections in the desired configuration (Appendix. A.4). The inset figure within Fig. 2 shows a full spectrum of difficulty levels found within our dataset, with over half of our dataset consisting of configurations with more than 4 interconnections, 4 resources, and over 42 LOC, and the longest configurations having either 280 LOC, 24 resources or 33 interconnections.

## 2.3 Infrastructure intent specifications

Just like regular coding problems, infrastructure problems can often be specified ambiguously, where the user's intention is not always clearly defined. To address this, we use OPA Rego to encode a range of possible correct configurations for a given problem, creating an infrastructure intent specification crafted by a human expert. This step is crucial because functional correctness processes alone, such as ensuring configurations can be compiled, cannot resolve ambiguities in the user's intent. Intent specs in our dataset vary in length, averaging 37.5 LOC, and the longest spec containing 205 LOC.

In general, the specification will contain three categories of intents (i.e., clarifying three sources of ambiguity): (1) valid resources, (2) optional attributes, and (3) required attributes. The valid resources specify their dependencies and the number of resources allowed. The optional attributes specify their existence, and/or the range of acceptable values whereas the required attributes mandate the range of acceptable values. Note that anything not included in the specification is considered incorrect. This is showcased via an example in Fig. 3(b): For example, for valid resources, the intent spec contains a validation block (`is_valid_aws_db_proxy`) that checks if an `aws_db_proxy` resource exists in the LLM-generated config. Further, specifying a connection timeout limit for an `aws_db_proxy` is optional according to cloud-provider guidelines; however, since it is explicitly specified in the prompt, the intent will enforce its value to be between 1800 and 3600 (while inferring that this is specified in seconds). As another example, the intent infers that daily DB backups are equivalent to simply including a `preferred_backup_window` attribute, with the example timing of morning backups being irrelevant. For required attributes, the `engine_family` attribute within the proxy is

**Prompt**: Set up an AWS Aurora cluster with a proxy for enhanced connection management. The proxy should close connections that have been inactive for a time (set this value to anywhere between 30 to 60 mins). I want the database to be some variant of SQL, that is not a Microsoft SQL Server. My DB should backup data every day (e.g., in the morning from 7 to 9).

```
resource "aws_db_proxy" "rds_proxy" {
  name                   = "test"
  engine_family          = "MYSQL"
  idle_client_timeout    = 2000
  role_arn               = aws_iam_role.proxy_role.arn

  auth {
    auth_scheme = "SECRETS"
    iam_auth    = "DISABLED"
    ....
  }
  ...
}

resource "aws_rds_cluster" "rds_cluster" {
  cluster_identifier      = "rds-cluster"
  engine                  = "aurora-mysql"
  engine_version          = "8.0.mysql_aurora.3.02.0"
  preferred_backup_window = "13:00-15:00"
  ...
}
```

**(a) Simplified correct config instance**

```
is_valid_aws_db_proxy {
    some i; resource := input.configuration.resources[i]
    resource.type == "aws_db_proxy"
    engine_set := {"MYSQL","POSTGRESQL"}
    engine_set[resource.engine_family]
    resource.idle_client_timeout ≤ 1800
    resource.idle_client_timeout ≥ 3600
}

is_valid_aws_rds_cluster {
    some i; resource := input.configuration.resources[i]
    resource.type == "aws_rds_cluster"
    is_valid_engine(resource.engine)
    resource.preferred_backup_window ≠ null
}

is_valid_aws_iam_role {
    some i; resource := input.configuration.resources[i]
    resource.type == "aws_iam_role"
    resource.assume_role_policy ≠ null
}
```

**(b) Simplified infra intent spec**

Figure 3: IaC-Eval dataset row simplified snippet.

constrained by the user prompt to only use a subset of valid cloud-provider-defined values. We do not need to verify the existence of required attributes, as these are automatically validated during the compilation phase.

We emphasize that crafting the initial batch of dataset intents is challenging, but there is significant reuse potential across multiple configurations. The intents we have created can serve as templates (e.g., a validation block for a given resource), allowing for easier adaptation with minor adjustments. Moreover, problems often imply the existence of many interlinked required or optional resources each with their own attributes. For instance, creating an EC2 instance not only requires an attached AWS network interface card but also depends on a VPC, which in turn requires subnets and route tables. This chain of dependencies can extend even further, including security groups, IAM roles, and policies. Since manually specifying all these details at this stage is impractical, we leave some aspects under-specified, similar to the approach seen in HumanEval [48]. An example is provided in Fig. 3(a), where an `IAM` role, which governs the actions executable by the DB proxy, is attached to the proxy via the `role_arn` attribute. The actual IAM role resource block itself is omitted for brevity. The `role_arn` is a required attribute of the resource, and its presence in a correct configuration is therefore implied (not explicitly stated) in the prompt. To simplify crafting intents, we only verify the existence of the `aws_iam_role` resource block and ensure that its policy is not empty; we do not check its content. With this base dataset, we envision a move towards the automatic synthesis of intents (Sec. 4) to facilitate the expansion of our dataset with increased service/provider coverage.

## 3 Experiments

### 3.1 Evaluating LLMs performance on IaC-Eval

We evaluate a range of code generation models used within the popular HumanEval [26] and EvalPlus [48] Python code benchmarks, against IaC-Eval. These include the top-ranked GPT4, WizardCoder [49] and Magicoder [70] models, and CodeLlama [50] variants. Inference was performed using OpenAI APIs for GPT-4 and GPT-3.5, and Replicate [60] endpoints for all the other models, except for Magicoder, which was deployed on a `g5.2xlarge` instance running an NVIDIA A10G GPU (24 GB memory) via AWS SageMaker. We use the unbiased version of pass@$k$ [26], a standard metric also used in the aforementioned benchmarks, to assess correctness by generating 20 samples for each problem. This method gives us the probability that at least one out of $k$ chosen samples among the 20 is correct. The IaC-Eval column of Table 1 showcases our results where scores are tabulated in terms of pass@1 accuracy: our top-ranked models are GPT-4, WizardCoder-33B-V1.1, and GPT3.5, achieving a score of 19.36%, 8.93%, and 7.99%, respectively. Our worst-performing model was CodeLlama instruct (7B) which scored 1.97%. This is in stark contrast to these models' performance on EvalPlus, where they scored 86.6%, 73.2%, 70.7%, and 35.4%, respectively; highlighting the

Table 1: Average benchmark scores for various models when tested against various evaluation metrics. Popular LLMs perform poorly on IaC-Eval, showcasing its difficulty.

| Model | | Evaluation metric | | | |
|---|---|---|---|---|---|
| Rank | Name | BLEU | CodeBERTScore | LLM-judge | IaC-Eval |
| 1 | GPT-4 | 18.49 | 83.39 | 61.79 | 19.36 |
| 2 | WizardCoder-33B-V1.1 | 15.22 | 80.50 | 28.72 | 8.93 |
| 3 | GPT-3.5-turbo | 14.52 | 77.26 | 34.49 | 7.99 |
| 4 | Magicoder-S-CL-7B | 14.22 | 79.49 | 23.14 | 7.62 |
| 5 | Gemini 1.0 Pro | 11.96 | 78.90 | 19.72 | 3.43 |
| 6 | CodeLlama Instruct (34B) | 11.47 | 78.64 | 11.97 | 2.99 |
| 7 | CodeLlama Instruct (13B) | 11.18 | 76.46 | 9.83 | 2.01 |
| 8 | CodeLlama Instruct (7B) | 9.31 | 70.22 | 7.18 | 1.97 |

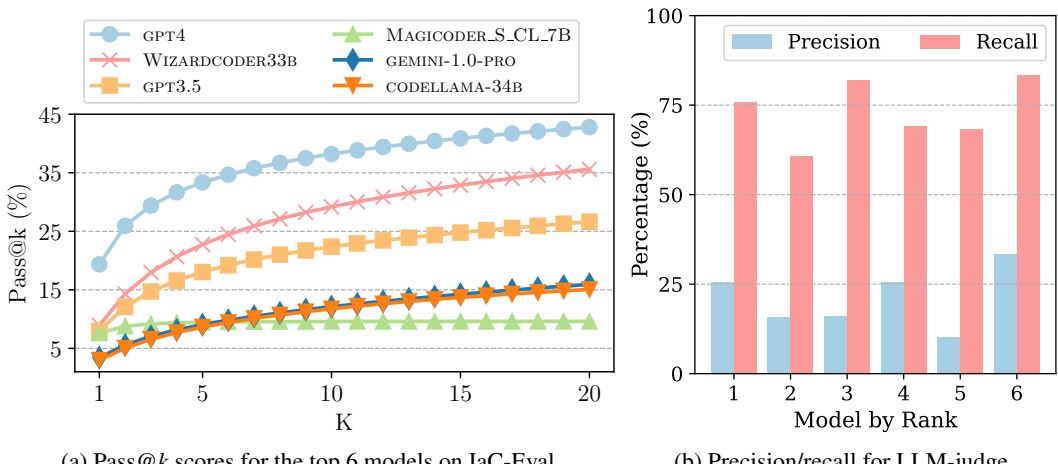

(a) Pass@$k$ scores for the top 6 models on IaC-Eval.

(b) Precision/recall for LLM-judge.

Figure 4: Multi-sample generation improves scores. LLM-judge decisions are unreliable.

difficulty of our dataset [32], and demonstrating that current models are ineffective at generating IaC code. Further, extended results containing pass@$k$ scores for all $k$ values are showcased for the Top-6 performing models in Fig. 4a. We observe improvements across all models with increasing $k$ values, except for Magicoder; upon closer inspection, we found that this is because Magicoder tends to get correct answers for a narrow set of problems consistently, where generating more samples does not yield significant accuracy improvements. Finally, we showcase an example incorrect generated IaC program in Fig. 6, demonstrating that even GPT-4 can exhibit issues such as hallucinating entire service components.

**Impact of intents and difficulty on evaluation scores.** To examine the utility of our benchmark's two-step pipeline, we compare the accuracy (Fig. 5a) when only Terraform compilation checks are used (blue hatched bars), and when both compilation and intent checks are used: across all models, intent specifications helped remove over ≈50% of false positives, i.e., programs deemed correct by the Terraform compilation phase but are actually incorrect as they do not satisfy user intent. Separately, our results in Fig. 5b show that complex IaC-Eval problems are harder for models to solve, demonstrating the efficacy of our system of difficulty levels.

## 3.2 Comparing IaC-Eval against baseline metrics

We compare IaC-Eval against three existing metrics with results tabulated in Table. 1:

**BLEU and CodeBERTScore.** Both are widely utilized to assess the quality of machine-produced translations [26, 73] by determining how similar some generated text is to a reference text; unlike BLEU [54] which is general purpose, CodeBERTScore [76] is specifically designed for code generation. Both produce scores ranging from 0 to 1, with higher values indicating better translation quality. Our results are displayed as percentages: though a downward trend in scores is observed for

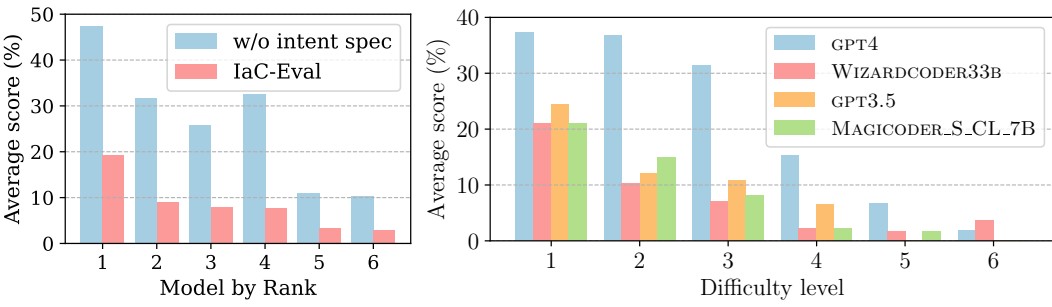

(a) Intent specs help filter false positives.  (b) Difficult questions in IaC-Eval are harder to solve.

Figure 5: Ablation: effectiveness of intent specifications and complexity levels.

both metrics, consistent with IaC-Eval, a deeper analysis reveals that these metrics cannot effectively differentiate between correct and incorrect solutions (as judged by IaC-Eval), as they frequently assign similar scores to both, with an average difference of only 6.9% for CodeBERTScore and 8.3% for BLEU. This finding is consistent with existing studies [73, 26], and motivates the use of functional correctness metrics instead.

**LLM-as-a-judge.** We also assess our models using the well-known LLM-as-a-judge metric, as referenced in previous studies [75, 46, 17]. For this evaluation, we employ GPT-4, the top-performing code model, as the judge. The downward trend in scores suggests that the LLM-judge has some intuition. However, Figure 4b reveals that its precision is low (hovering between 10% and 30%), indicating a high rate of misclassification where incorrect solutions are frequently judged as correct.

### 3.3 Assessing common enhancement strategies

**Few-shot and chain-of-thought.** These prompting strategies are commonly used in code generation tasks [24, 40]. Few-shot prompting involves providing a few examples to guide the model. The specific prompt used in our experiments is detailed in Appendix. B.1, where we employ three-shot prompting [73]. Chain-of-thought reasoning (CoT) guides the model through a step-by-step process to arrive at a solution, mimicking human logical progression. In our approach, we ask the model to reason through the overall structure of the desired infrastructure, covering services, resources, and individual attributes (Appendix. B.2). Our findings indicate that these methods yield unreliable improvements, sometimes even degrading performance. For instance, both strategies reduced GPT-4's score from 19.36% to 10.64% and 9.31%, respectively. This is likely due to the extensive service diversity within cloud infrastructure, limiting the effectiveness of these strategies.

**Multi-turn.** This is a prompting strategy widely used in existing work [62, 71]. Here we assume the model has access to an interpreter to evaluate the generation; errors within the compilation phase are passed back to the LLM as additional context for program repair. Our setup involves two turns of interaction between the model and the interpreter. We observe a noticeable improvement across our Top-4 models (5.5% on average), and a slight decrease in performance for our remaining models (0.92% on average). The specific prompt used is detailed in Appendix. B.3.

**Retrieval-augmented generation (RAG) [63].** This approach leverages a vector database [47] that stores relevant IaC documentation [37] for contextual retrieval. Given the problem input, the LLM first helps identify key terms and contextual cues to retrieve relevant documentation and code snippets from our database. The LLM then uses the retrieved documentation snippets as context to generate the final IaC code. This approach allows the LLM to access examples of similar code snippets, API documentation, or coding patterns pertinent to the problem, enabling more accurate and contextually appropriate generation. We observe a noticeable improvement across all models, with an average increase of 6.14%, making this the most effective strategy we've evaluated. Our prompt used is shown in Appendix. B.4.

Overall, these methods may provide modest improvements, but we hope the introduction of IaC-Eval will catalyze further research specifically tailored to improving IaC code generation.

Table 2: Average benchmark scores of various models when enhanced with differing strategies, ranked by highest score across all strategies. Evaluation performed against IaC-Eval. Performance is slightly improved across the board, especially with multi-turn and RAG.

| Model | | Enhancement strategy | | | |
|---|---|---|---|---|---|
| Rank | Name | Few-shot | CoT | Multi-turn | RAG |
| 1 | GPT-4 | 10.64 | 9.31 | 31.12 | 36.70 |
| 2 | GPT-3.5-turbo | 0.80 | 1.60 | 11.44 | 21.81 |
| 3 | Magicoder-S-CL-7B | 2.93 | 0.53 | 12.50 | 12.77 |
| 4 | WizardCoder-33B-V1.1 | 1.60 | 1.06 | 9.04 | 11.70 |
| 5 | CodeLlama Instruct (34B) | 3.19 | 3.19 | 2.13 | 6.12 |
| 6 | CodeLlama Instruct (7B) | 2.39 | 3.72 | 0.53 | 5.59 |
| 7 | Gemini 1.0 Pro | 1.33 | 0.00 | 2.93 | 5.32 |
| 8 | CodeLlama Instruct (13B) | 1.06 | 1.86 | 1.06 | 3.46 |

**Prompt**: Create a Lambda Layer Version resource from "lambda_layer_payload.zip"

```
resource "aws_lambda_layer_version" "lambda_layer" {
  filename   = "lambda_layer_payload.zip"
  layer_name = "lambda_layer_name"

  compatible_runtimes = ["nodejs16.x"]
}
```

**(a) Simplified correct config instance**

```
resource "aws_lambda_layer" "example" {
  layer_name = "my-lambda-layer"
  source_code_hash = filebase64sha256("lambda_layer_payload.zip")
  compatible_runtimes = ["nodejs14.x", "python3.8", "java11"]
  description = "My Lambda Layer Description"
}

resource "aws_lambda_layer_version" "example" {
  layer_name = aws_lambda_layer.example.layer_name
  description = "Version 1.0"
  compatible_runtimes = aws_lambda_layer.example.compatible_runtimes
  s3_bucket = aws_lambda_layer.example.source_code_s3_bucket
  s3_key = aws_lambda_layer.example.source_code_s3_key
}
```

**Hallucinated entire resource block and all attribute params/values**

**Incorrect usage: expects a string, not reference**

**Referencing non-existing resource**

**(b) Incorrect GPT-4 generated config**

Figure 6: Example errors from a GPT-4 generated configuration.

# 4 Discussion

**Future directions.** One approach is to develop more intricate evaluation metrics to address specific issues like security breaches, policy adherence, and code quality aspects such as brevity, clarity, and style [38]. The aim could be to demonstrate that existing models lack "security-awareness" and could lead to problems, such as: (1) vulnerability to adversarial information extraction attacks, where attackers can extract IaC-specific secrets from LLMs [25], or (2) failure to follow security best practices [27], potentially causing naive users to push unsafe code into production. Second, given our base dataset, we could expand it by implementing automated dataset creation for fine-tuning or evaluation purposes [48]. Third, advanced compile-time checks are an existing line of work [15, 2, 59] that we could integrate to increase our coverage. Fourth, IaC-Eval currently focuses on Terraform, the most popular IaC tool, but integrating other frameworks such as AWS CloudFormation and Pulumi are planned for future work (e.g., since they share similar intermediate representations as Terraform [58]). Finally, in terms of improving IaC code generation, one potential avenue is utilizing two-level synthesis or context injection, which could tailor the code more accurately to specific contexts, enhancing personalization and precision.

**Limitations.** There are several avenues for expansion. IaC-Eval currently does not include function generation, although supported by HCL. Additionally, IaC-Eval focuses on AWS, but we plan to extend support to other cloud providers like Azure. Despite these limitations, the core concepts of

our benchmark (e.g., intent specifications), are applicable across other cloud platforms such as Azure and Google Cloud because IaC is built to be cloud-provider agnostic. As this is the first benchmark and dataset focused on IaC, we chose AWS as it is the most popular cloud provider, and is broadly adopted by the community. Our dataset focuses on AWS services because creating these datasets is time-intensive and requires an in-depth understanding of each cloud service in concern, especially since we cover a wide range of services and include questions of varying difficulty levels.

**Data collection and annotation process.** Our team learned from official AWS and IaC documentation, and other learning materials (including a mixture of public IaC repositories within GitHub, and IaC-related StackOverflow/Reddit posts). All configurations were tested to ensure correctness. Additionally, our team includes experts who have been using IaC for years. To maintain high quality, we have implemented channels (GitHub/HuggingFace issue trackers) for the community to report any issues/errors, which we will promptly address–this is explicitly mentioned in our supplementary material and repositories. Finally, we also welcome contributions from the community. More details are available within Section C3 of our supplementary material.

**Potential societal impact.** We do not foresee any potential negative societal impacts arising from our work on IaC-Eval. While IaC-Eval does evaluate LLM-generated IaC programs, it's important to note that these evaluations are limited to compile-time operations and do not involve deploying the programs into the cloud. This approach ensures that users need not worry about any potential impact on their cloud infrastructure. Additionally, we have conducted thorough manual reviews of our dataset to confirm that it is safe for execution, posing no risk to either cloud providers or users. Moreover, the composition, collection process, and other relevant details of our dataset are provided in our supplementary material, which adheres to the "datasheets for datasets" format to ensure transparency and accountability.

## 5 Related work

**LLMs for code.** LLMs have shown impressive performance in code generation (and its subtasks including code translation [42, 61] and program repair [72]) for general-purpose programming languages, where a model is typically provided with natural language descriptions of the desired functionality and asked to generate the corresponding code snippet. Models include OpenAI's ChatGPT/GPT-4, Codex [52], CodeLlama [50], and WizardCoder [49]. Our work aims to determine, for the first time, whether these models can also perform well in generating IaC programs (Sec. 3).

**Code evaluation metrics for LLMs.** Text similarity metrics (e.g., BLEU [54]) and human evaluation [44] do not work well for evaluating general-purpose programming languages for various reasons (e.g., minor dissimilarities can result in significant problems/errors [19], while human evaluation is not scalable). LLM code generation is instead commonly evaluated based on functional correctness using the pass@$k$ [26] metric, which makes use of input-output test cases: a code segment is considered correct if it passes all such test cases. IaC-Eval focuses on functional correctness as well, but applies this to a cloud infrastructure scenario using intent specifications.

**Coding benchmarks for LLMs.** These include the widely studied HumanEval [26] benchmark consisting of 164 hand-crafted programming problems, and the MBPP [19] (1K programming problems) benchmark, built for evaluating Python code generation. Benchmarks targeting other languages include Spider [74] (SQL), CodeContests [45] (C++ and Java). CloudEval-YAML [73] is closely related to our work as it focuses on the cloud domain; it provides a dataset for specific cloud applications (i.e., Kubernetes, Envoy, and Istio) composed of simple YAML configurations. IaC-Eval instead focuses on cloud infrastructure, a vastly different and more complex domain, and introduces the first IaC dataset consisting of 458 hand-crafted problems, and a unique evaluation pipeline (Sec. 2).

**IaC frameworks.** IaC frameworks simplify cloud infrastructure development by codifying resources to ensure consistency and repeatability. Similar paradigms exist, such as Kubernetes, a container orchestration platform which abstracts away the complexities of managing containers, and serverless computing, which allows developers to run/deploy applications without server management. Apart from Terraform, other cloud-agnostic IaC frameworks include Pulumi [12], OpenTofu [11], and Crossplane [5], as well as cloud-specific tools such as Azure Bicep and AWS CloudFormation. Preceding these frameworks, we have Ansible [1], Chef [3], and Puppet [13], which are mainly used for in-system configuration and automation of software/application setups, on individual systems

that may be part of a larger network/computing infrastructure. Cloud infrastructure can also be provisioned using non-IaC cloud-level APIs as well (e.g., through SDKs such as AWS Boto3), as is done in some existing works [41], though this is a much harder process to navigate [58].

**IaC tools.** There are a broad range of industry tools designed to enhance specific aspects of IaC management, such as security checkers (e.g., TFSec [16], Terrascan [15] and KICS [10]), and drift detection tools (e.g., driftctl [6]). Additionally, various policy engines such as OPA Rego [51], the HashiCorp Sentinel framework [8], and Cloud Custodian [4] help developers define and enforce compliance rules within IaC programs. Recently, we have also seen the gradual introduction of LLMs for IaC program generation, with tools like Pulumi AI [57], Firefly [7], and InfraCopilot [9]. Finally, academic projects have focused on directions such as enhanced IaC program validation [58], the cloudless computing vision project [59], fault analysis [30], and vulnerability detection [43].

## 6  Conclusion

We introduce IaC-Eval, the first dataset and benchmark capable of evaluating IaC code generation by LLMs. IaC-Eval comprises 458 human-curated scenarios that cover a diverse range of popular AWS services and varying difficulty levels. Our evaluation reveals that current LLMs, including GPT-4, perform poorly on IaC-Eval, with a pass@1 accuracy of 19.36%, compared to 86.6% on the EvalPlus Python benchmark. This underscores the need for advancements in LLM-based IaC code generation. We open-source IaC-Eval to facilitate future research in this domain.

## 7  Acknowledgments

We thank the anonymous reviewers for their insightful feedback. This work was partially supported by a Cisco Research Grant, AWS Cloud Research Credits, a VMware Early Career Faculty Grant, as well as NSF grants CNS-1942219, CNS-2106751, CNS-2106388, and CNS-2214272.

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

Table 3: IaC-Eval dataset columns

| Column name | Descriptor |
|---|---|
| Resource | All resources that this scenario requires. |
| Prompt | User question to be fed to the LLM under evaluation. |
| Rego intent | Validates the generated program for user intent fulfilment. |
| Difficulty | Calculated difficulty level (Table. 4). |
| Intent | Natural language intent for easy viewing. |
| Reference output | Example correct program. |

# A    Additional IaC-Eval details

## A.1    Dataset URL and links

**Dataset.** Links to download or view our dataset can be found on our GitHub homepage at hhttps://github.com/autoiac-project/iac-eval. The dataset's homepage, including all future versions and updates, is hosted at https://huggingface.co/datasets/autoiac-project/iac-eval.

**Dataset Documentation.** We offer comprehensive descriptions for the dataset and its usage through the HugggingFace dataset link above. The page covers the data layout, essential metadata, and ways to load the data for usage.

**Dataset DOI.** Our persistent DOI is https://doi.org/10.57967/hf/2400.

**Code.** Our code is available at hhttps://github.com/autoiac-project/iac-eval.

**Code Documentation.** Within our GitHub repository above, we provide detailed descriptions of our evaluation benchmark, and instructions on how to setup the benchmark, and run each step to reproduce our evaluation results; along with instructions on setting up the LLMs required: e.g., on acquiring the required API keys or setting up of AWS SageMaker.

## A.2    Data format

Our dataset (located in our HuggingFace repository) is in a standard CSV format. There is currently only one file, where we store all our AWS scenarios. The file has the following columns: "Resource", "Prompt", "Intent", "Rego intent", "Difficulty", and "Reference output". An explanation of their intended use is provided in Table. 3.

## A.3    Maintenance and long-term preservation

The authors of IaC-Eval are committed to the ongoing maintenance and preservation of the dataset, with plans to potentially expand it for future research endeavors. This commitment includes addressing any issues identified by the community after the dataset's release, with feedback monitored through GitHub or HuggingFace issue trackers. All data is hosted on GitHub and HuggingFace to ensure reliable and stable storage. Further, we may also consider migrating/replicating the data to archival storage for long-term preservation.

**Findable.** Our dataset is stored in the HuggingFace repository listed in the previous subsection. All present and future data will share a global and persistent DOI https://doi.org/10.57967/hf/2400.

**Accessible.** Our code can be found through our GitHub repository listed above. Additionally, all data and descriptive metadata regarding our dataset can be downloaded from the public links listed on the GitHub repository, or the HuggingFace repository link listed above.

**Interoperable.** Our dataset is stored in standard CSV files that can be read using many common libraries, such as Pandas for Python.

**Reusable.** IaC-Eval's dataset is released under a permissive Creative Commons Attribution 4.0 license, while our code is released under a permissive MIT license.

## A.4    IaC-Eval difficulty levels

Details in Table. 4.

Table 4: IaC-Eval difficulty levels

| Difficulty level | Descriptor |
|---|---|
| 1 | LOC < 10 and Resource count < 2 and Interconnections count < 2 |
| 2 | LOC < 20 and Resource count < 4 and Interconnections count < 4 |
| 3 | LOC < 40 and Resource count < 6 and Interconnections count < 6 |
| 4 | LOC < 60 and Resource count < 8 and Interconnections count < 8 |
| 5 | LOC < 80 and Resource count < 10 and Interconnections count < 10 |
| 6 | LOC $\geq$ 80 or Resource count $\geq$ 10 or Interconnections count $\geq$ 10 |

# B  Experimental setup details

Regarding full details of all our prompts used, please refer to our GitHub repository.

## B.1  Few-shot prompt template

```
[System prompt]
You are TerraformAI, an AI agent that builds and deploys Cloud
Infrastructure written in Terraform HCL. Generate a description of the
Terraform program you will define, followed by a single Terraform HCL
program in response to each of my Instructions. Make sure the
configuration is deployable. Create IAM roles as needed. If variables
are used, make sure default values are supplied. Be sure to include a
valid provider configuration within a valid region. Make sure there are
no undeclared resources (e.g., as references) or variables, i.e., all
resources and variables needed in the configuration should be fully
specified.

Here are a few examples:

Example prompt 1: {prompt-1}
Example output 1:
'''hcl
{output-1}
'''

Example prompt 2: {prompt-2}
Example output 2:
'''hcl
{output-2}
'''

Example prompt 3: {prompt-3}
Example output 3:
'''hcl
{output-3}
'''

Here is the actual question to answer:
{question}
```

## B.2 Chain-of-thought prompt template

```
[System prompt]
You are TerraformAI, an AI agent that builds and deploys Cloud
Infrastructure written in Terraform HCL. Generate a description of the
Terraform program you will define, followed by a single Terraform HCL
program in response to each of my Instructions. Make sure the
configuration is deployable. Create IAM roles as needed. If variables
are used, make sure default values are supplied. Be sure to include a
valid provider configuration within a valid region. Make sure there are
no undeclared resources (e.g., as references) or variables, i.e., all
resources and variables needed in the configuration should be fully
specified.

Here are a few examples:

Example prompt 1: Create an AWS RDS instance (with an instance class of
db.t2.micro, and don't create a final snapshot before eventual deletion)
with randomly generated id and password.
Example output 1: Let's think step by step. First, let's reason about
the resources needed: this would be an AWS RDS instance
(aws_db_instance), and resources to generate a random id and password.
Second, we fill in the attributes of each resource, starting with those
explicitly and implicitly mentioned in the prompt, and followed by
others: for example, for the aws_db_instance, we need to set the
"instance_class" attribute to "db.t2.micro", and the
"skip_final_snapshot" attribute to true. Finally, we connect the
resources together, as needed: here "identifier" should be connected to
the "random_id" resource, and "password" should be connected to the
"random_password" resource.
'''hcl
{output-1}
'''

Example prompt 2: {prompt-2}
Example output 2: {CoT-output-2}
'''hcl
{output-2}
'''

Example prompt 3: {prompt-3}
Example output 3: {CoT-output-3}
'''hcl
{output-3}
'''

Here is the actual question to answer:
{question}
```

## B.3 Multi-turn prompt template

This prompt is used if a Terraform plan error was observed in the first turn:

```
[System prompt]
You are TerraformAI, an AI agent that builds and deploys Cloud
Infrastructure written in Terraform HCL. Given an incorrect Terraform
program along with an error message, your task is to first describe the
error in your own words, followed by a description of the fix you will
apply, and ending with a single corrected Terraform HCL program. Make
sure the configuration is deployable. Create IAM roles as needed. If
variables are used, make sure default values are supplied. Be sure to
include a valid provider configuration within a valid region. Make sure
there are no undeclared resources (e.g., as references) or variables,
that is, all resources and variables needed in the configuration should
be fully specified.

Here is the original prompt:
{prompt}

Here is the incorrect configuration:
{generated-config}

Here is the Terraform plan error message:
{error-message}
```

This prompt is used if an intent specification error is observed in the first turn:

```
[System prompt]
You are TerraformAI, an AI agent that builds and deploys Cloud
Infrastructure written in Terraform HCL. Given an incorrect Terraform
program along with an error message, your task is to first describe the
error in your own words, followed by a description of the fix you will
apply, and ending with a single corrected Terraform HCL program. Make
sure the configuration is deployable. Create IAM roles as needed. If
variables are used, make sure default values are supplied. Be sure to
include a valid provider configuration within a valid region. Make sure
there are no undeclared resources (e.g., as references) or variables,
that is, all resources and variables needed in the configuration should
be fully specified.

Here is the original prompt:
{prompt}

Here is the incorrect configuration:
{generated-config}

Here is the Rego OPA policy associated with this configuration:
{intent-spec}

Here is the Rego OPA policy error message:
{error-message}
```

## B.4 RAG prompt template

```
[System prompt]
You are TerraformAI, an AI agent that builds and deploys Cloud
Infrastructure written in Terraform HCL. Generate a description of the
Terraform program you will define, followed by a single Terraform HCL
program in response to each of my Instructions. Make sure the
configuration is deployable. Create IAM roles as needed. If variables
are used, make sure default values are supplied. Be sure to include a
valid provider configuration within a valid region. Make sure there are
no undeclared resources (e.g., as references) or variables, i.e., all
resources and variables needed in the configuration should be fully
specified.

Here is some additional knowledge/context retrieved from Terraform
documentation, that may (or may not) potentially help you answer the
question:
{context}

Here is the actual question to answer:
{question}
```

## B.5 Detailed experimental setup

Table 5: Overview of models evaluated by IaC-Eval.

| Model Name | Size | Release Year | Open-Source |
|---|---|---|---|
| GPT-4 [53] | N/A | 2023 | |
| GPT-3.5-turbo [53] | N/A | 2022 | |
| Magicoder-S-CL-7B [70] | 7B | 2023 | ✓ |
| WizardCoder-33B-V1.1 [49] | 33B | 2023 | ✓ |
| Gemini 1.0 Pro [29] | N/A | 2023 | |
| CodeLlama-7B [50] | 7B | 2023 | ✓ |
| CodeLlama-13B [50] | 13B | 2023 | ✓ |
| CodeLlama-34B [50] | 34B | 2023 | ✓ |

We evaluated a range of state-of-the-art models as detailed in Table. 5, which provides a summary of the models evaluated, their sizes (model parameters in billions), the Release Year indicating when the model was launched, and the Open-Source column highlighting models with publicly available weights. In total, we evaluated 8 prominent LLMs that have also been evaluated in the popular HumanEval [26] and EvalPlus [48] code generation benchmarks. For each model, we randomly sample 20 programs. Both GPT and our Gemini models were evaluated using their default settings: greedy decoding where the temperature was set to 0, and unset max token lengths. All the other models were evaluated using a temperature of 0.6, and 512 max tokens. Inference was performed using OpenAI APIs for GPT-4 and GPT-3.5, and Replicate [60] endpoints for all the other models, except for Magicoder, which was deployed on a `g5.2xlarge` instance running an NVIDIA A10G GPU (24 GB memory) via AWS SageMaker. We adhered to the official examples provided for each model (e.g., on HuggingFace model cards) to design their prompts. Specifically, the prompts are instruction-based, incorporating simple instructions to frame the user prompt, thereby explicitly guiding the LLMs towards code generation. The code blocks for each model's output was extracted using a set of well-known and observed delimiters (e.g., within triple backtick blocks). For our RAG experimental setup, we leveraged the LlamaIndex vector database [47], using it to store all relevant IaC documentation [37] for contextual retrieval. We chunked the data into 500-word segments, each indexed separately to enhance retrieval speed and accuracy by ensuring that search operations are confined to the most relevant subsets of data. For each user prompt, we fed it into GPT-3.5-turbo which acted as an intermediary LLM, to summarize and convert the user prompt into a set of questions for more effective retrieval.

