# C  IaC-Eval datasheet

We provide details on the IaC-Eval dataset published on HuggingFace, with a persistent DOI at https://doi.org/10.57967/hf/2400. Our Croissant metadata can be found in the same HuggingFace repository above. Additional information, and our evaluation benchmark can be found in our GitHub repository at https://github.com/autoiac-project/iac-eval.

## C.1  Motivation

1. **For what purpose was the dataset created?** Was there a specific task in mind? Was there a specific gap that needed to be filled? Please provide a description.

   The dataset and benchmark was created to enable the quantitative evaluation of large language model performance on generating Infrastructure-as-Code (IaC) programs. There currently exist many such datasets/benchmarks (e.g., HumanEval) that evaluate general code generation (e.g., Python code generation for HumanEval), but none are capable of evaluating IaC code generation. This enables future research in developing IaC-specific code generation techniques: we can use IaC-Eval to determine how effective these techniques are.

2. **Who created this dataset (e.g. which team, research group) and on behalf of which entity (e.g., company, institution, organization)?**

   This dataset was primarily created by the authors from the University of Michigan, in collaboration with Myungjin Lee from Cisco Research.

3. **Who funded the creation of the dataset?** If there is an associated grant, please provide the name of the grant or and the grant name and number.)

   This work is partially funded by Cisco and Amazon.

4. **Any other comments?**

   No.

## C.2  Composition

1. **What do the instances that comprise the dataset represent (e.g. documents, photos, people, countries)?** Are there multiple types of instances (e.g. movies, users, and ratings; people and interactions between them; nodes and edges)? Please provide a description.)

   Each instance (row) of the dataset represents an IaC problem scenario that we use to evaluate LLMs on their IaC code generation capabilities. Each scenario essentially contains a user's natural language IaC problem description (e.g., creating an AWS compute node) and its associated infrastructure intent specification (that codifies precisely what the user's intention is). The former is fed to the LLM under evaluation, while the latter is used to verify that the LLM's generated program conforms to the user's intent. Our scenarios cover a variety of AWS services, and range from simple to challenging.

2. **How many instances are there in total (of each type, if appropriate)?**

   We have 458 human-curated scenarios, with details provided in Sec. 3.2.

3. **Does the dataset contain all possible instances or is it a sample(not necessarily random) of instances from a larger set?** If the dataset is a sample, then what is the larger set? Is the sample representative of the larger set (e.g. geographic coverage)? If so, please describe how this representativeness was validated/verified. If it is not representative of the larger set, please describe why not (e.g., to cover a more diverse range of instances, because instances were withheld or unavailable).)

   IaC-Eval is the first step in this research direction, and we cover a wide range of popular services within AWS. However, as alluded to in Sec. 1, the diversity of cloud services and providers makes it impractical, especially in the first work, to cover all possible scenarios. We hope our work will inspire and lead to future expansions to the dataset, whether it be human-curated or automated expansions.

4. **What data does each instance consist of?** ("Raw" data (e.g. unprocessed text or images) or features? In either case, please provide a description.)

Each row in our dataset makes up an individual instance. The format of each row is described in Sec. C.10.

5. **Is there a label or target associated with each instance?** If so, please provide a description.

Each instance contains an infrastructure intent specification (as mentioned earlier) and an example reference output (which we envision could be useful in the future for fine-tuning purposes).

6. **Is any information missing from individual instances?** (If so, please provide a description, explaining why this information is missing (e.g., because it was unavailable). This does not include intentionally removed information, but might include, e.g., redacted text.)

No.

7. **Are relationships between individual instances made explicit (e.g., users' movie ratings, social network links)?** (If so, please describe how these relationships are made explicit.)

The instances are independent of each other.

8. **Are there recommended data splits (e.g., training, development/validation, testing)?** (If so, please provide a description of these splits, explaining the rationale behind them.)

We leave the training, validation and testing splits to the discretion of the users themselves, depending on the downstream task the dataset is used for. With that being said, we currently envision the base dataset of IaC-Eval to be used for evaluation purposes, in which case all of the existing dataset would be used as a testing split.

9. **Are there any errors, sources of noise, or redundancies in the dataset?** (If so, please provide a description.)

There could potentially be errors in our dataset. However, we are committed to the ongoing maintenance and preservation of the dataset, and this commitment includes addressing issues identified by the community after the dataset's release, with feedback monitored through GitHub or HuggingFace issue trackers.

10. **Is the dataset self-contained, or does it link to or otherwise rely on external resources (e.g., websites, tweets, other datasets)?** (If it links to or relies on external resources, a) are there guarantees that they will exist, and remain constant, over time; b) are there official archival versions of the complete dataset (i.e., including the external resources as they existed at the time the dataset was created); c) are there any restrictions (e.g., licenses, fees) associated with any of the external resources that might apply to a future user? Please provide descriptions of all external resources and any restrictions associated with them, as well as links or other access points, as appropriate.)

Yes.

11. **Does the dataset contain data that might be considered confidential (e.g., data that is protected by legal privilege or by doctor-patient confidentiality, data that includes the content of individuals' non-public communications)?** (If so, please provide a description.)

No.

12. **Does the dataset contain data that, if viewed directly, might be offensive, insulting, threatening, or might otherwise cause anxiety?** (If so, please describe why.)

No.

13. **Does the dataset relate to people?** (If not, you may skip the remaining questions in this section.)

No.

14. **Does the dataset identify any subpopulations (e.g., by age, gender)?** (If so, please describe how these sub-populations are identified and provide a description of their respective distributions within the dataset.)

N/A.

15. **Is it possible to identify individuals (i.e., one or more natural persons), either directly or indirectly (i.e., in combination with other data) from the dataset?** (If so, please describe how.)

N/A.

16. **Does the dataset contain data that might be considered sensitive in any way (e.g., data that reveals racial or ethnic origins, sexual orientations, religious beliefs, political opinions or union memberships, or locations; financial or health data; biometric or genetic data; forms of government identification, such as social security numbers; criminal history)?** (If so, please provide a description.)

N/A.

17. **Any other comments?**

No.

## C.3 Collection process

1. **How was the data associated with each instance acquired?** (Was the data directly observable (e.g., raw text, movie ratings), reported by subjects (e.g., survey responses), or indirectly inferred/derived from other data (e.g., part-of-speech tags, model-based guesses for age or language)? If data was reported by subjects or indirectly inferred/derived from other data, was the data validated/verified? If so, please describe how.)

Our data was collected by studying the associated cloud services and their use cases (e.g., reading relevant documentations) and conducting deployments to ensure the scenarios were sound.

2. **What mechanisms or procedures were used to collect the data (e.g.,hardware apparatus or sensor, manual human curation, software program, software API)?** (How were these mechanisms or procedures validated?)

Our scenarios were crafted using deployments on an AWS account managed by our research group at the University of Michigan.

3. **If the dataset is a sample from a larger set, what was the sampling strategy (e.g., deterministic, probabilistic with specific sampling probabilities)?**

The dataset scenarios were selected to capture a wide variety of popular AWS services, since it is impractical to capture all possible cloud services from all cloud providers.

4. **Who was involved in the data collection process (e.g., students, crowd workers, contractors) and how were they compensated (e.g., how much were crowd workers paid)?**

Data collection was done by the authors.

5. **Over what timeframe was the data collected?** (Does this timeframe match the creation timeframe of the data associated with the instances (e.g., recent crawl of old news articles)? If not, please describe the timeframe in which the data associated with the instances was created.)

The dataset was created over the time period Sep 2023-May 2024.

6. **Were any ethical review processes conducted (e.g., by an institutional review board)?** (If so, please provide a description of these review processes, including the outcomes, as well as a link or other access point to any supporting documentation.)

No.

7. **Does the dataset relate to people?** (If not, you may skip the remaining questions in this section.)

No.

8. **Did you collect the data from the individuals in question directly, or obtain it via third parties or other sources (e.g., websites)?**

N/A.

9. **Were the individuals in question notified about the data collection?** (If so, please describe (or show with screenshots or other information) how notice was provided, and provide a link or other access point to, or otherwise reproduce, the exact language of the notification itself.)

N/A.

10. **Did the individuals in question consent to the collection and use of their data?** (If so, please describe (or show with screenshots or other information) how consent was requested and provided, and provide a link or other access point to, or otherwise reproduce, the exact language to which the individuals consented.)

N/A.

11. **If consent was obtained, were the consenting individuals provided with a mechanism to revoke their consent in the future or for certain uses?** (If so, please provide a description, as well as a link or other access point to the mechanism (if appropriate).)

N/A.

12. **Has an analysis of the potential impact of the dataset and its use on data subjects (e.g., a data protection impact analysis) been conducted?** (If so, please provide a description of this analysis, including the outcomes, as well as a link or other access point to any supporting documentation.)

N/A.

13. **Any other comments?**

No.

## C.4 Preprocessing/cleaning/labeling

1. **Was any preprocessing/cleaning/labeling of the data done(e.g.,discretization or bucketing, tokenization, part-of-speech tagging, SIFT feature extraction, removal of instances, processing of missing values)?** (If so, please provide a description. If not, you may skip the remainder of the questions in this section.)

Labelling of the resource composition and difficulty levels for each scenario was done automatically using a script, with the assigned value guided by string matching for the former, and Table. A.4 for the latter. We also did multiple rounds of manual inspection of our data to ensure they were sound.

2. **Was the "raw" data saved in addition to the preprocessed/cleaned/labeled data (e.g., to support unanticipated future uses)?** (If so, please provide a link or other access point to the "raw" data.)

Since our processing of the data left the raw data untouched (i.e., we only added additional columns), our raw data is present in our finalized dataset.

3. **Is the software used to preprocess/clean/label the instances available? (If so, please provide a link or other access point.)**

Our labelling scripts are included in our GitHub repository: https://github.com/autoiac-project/iac-eval

4. **Any other comments?**

No.

## C.5 Uses

1. **Has the dataset been used for any tasks already?** (If so, please provide a description.)

Not at the moment.

2. **Is there a repository that links to any or all papers or systems that use the dataset?** (If so, please provide a link or other access point.)

Not currently.

3. **What (other) tasks could the dataset be used for?**

   Apart from our envisioned evaluation use-case, we believe an expanded dataset could also be used for fine-tuning purposes. Finally, we welcome other innovative use-cases by the community!

4. **Is there anything about the composition of the dataset or the way it was collected and preprocessed/cleaned/labeled that might impact future uses?** (For example, is there anything that a future user might need to know to avoid uses that could result in unfair treatment of individuals or groups (e.g., stereotyping, quality of service issues) or other undesirable harms (e.g., financial harms, legal risks) If so, please provide a description. Is there anything a future user could do to mitigate these undesirable harms?)

   No.

5. **Are there tasks for which the dataset should not be used?** (If so, please provide a description.)

   No.

6. **Any other comments?**

   No.

## C.6 Distribution

1. **Will the dataset be distributed to third parties outside of the entity (e.g., company, institution, organization) on behalf of which the dataset was created?** (If so, please provide a description.)

   Yes, the dataset is freely and publicly available and accessible.

2. **How will the dataset will be distributed (e.g., tarball on website, API, GitHub)?** (Does the dataset have a digital object identifier (DOI)?)

   The dataset is free for download by everyone. Links are available in the GitHub repository: https://github.com/autoiac-project/iac-eval. The persistent DOI of the dataset is https://doi.org/10.57967/hf/2400.

3. **When will the dataset be distributed?**

   The first version of the dataset is distributed as of June 2024.

4. **Will the dataset be distributed under a copyright or other intellectual property (IP) license, and/or under applicable terms of use (ToU)?** (If so, please describe this license and/or ToU, and provide a link or other access point to, or otherwise reproduce, any relevant licensing terms or ToU, as well as any fees associated with these restrictions.)

   The dataset and our code are licensed under a CC-BY-4.0 and MIT license respectively.

5. **Have any third parties imposed IP-based or other restrictions on the data associated with the instances?** (If so, please describe these restrictions, and provide a link or other access point to, or otherwise reproduce, any relevant licensing terms, as well as any fees associated with these restrictions.)

   No.

6. **Do any export controls or other regulatory restrictions apply to the dataset or to individual instances?** (If so, please describe these restrictions, and provide a link or other access point to, or otherwise reproduce, any supporting documentation.)

   No.

7. **Any other comments?**

   No.

## C.7 Maintenance

1. **Who is supporting/hosting/maintaining the dataset?**

The dataset is maintained by the research groups associated with the authors from the University of Michigan.

2. **How can the owner/curator/manager of the dataset be contacted (e.g., email address)?**

   The manager of the dataset can be reached at `patkon@umich.edu`.

3. **Is there an erratum?** (If so, please provide a link or other access point.)

   There is no erratum currently. However, if errors are encountered, a fresh version will be released at the same repositories aforementioned. The repository issue trackers will contain a history of such updates.

4. **Will the dataset be updated (e.g., to correct labeling errors, add new instances, delete instances')?** (If so, please describe how often, by whom, and how updates will be communicated to users (e.g., mailing list, GitHub)?)

   Same as above.

5. **If the dataset relates to people, are there applicable limits on the retention of the data associated with the instances (e.g., were individuals in question told that their data would be retained for a fixed period of time and then deleted)?** (If so, please describe these limits and explain how they will be enforced.)

   N/A.

6. **Will older versions of the dataset continue to be supported/hosted/maintained?** (If so, please describe how. If not, please describe how its obsolescence will be communicated to users.)

   Versioning of the dataset and benchmark will be maintained in the repositories.

7. **If others want to extend/augment/build on/contribute to the dataset, is there a mechanism for them to do so?** (If so, please provide a description. Will these contributions be validated/verified? If so, please describe how. If not, why not? Is there a process for communicating/distributing these contributions to other users? If so, please provide a description.)

   IaC-Eval is publicly available and anyone with the available compute resources, and an AWS account should be able to extend to our dataset. We welcome contributions from the community, as IaC-Eval is merely a first step in this direction!

8. **Any other comments?**

   No.

## C.8   Reproducibility of IaC-Eval evaluation results

The evaluation setup details and step-by-step instructions on reproducing our results are found in IaC-Eval's GitHub repository: https://github.com/autoiac-project/iac-eval.

## C.9   Reading and using the dataset

Our dataset (located in our HuggingFace repository) is in a standard CSV format. There is currently only one file, where we store all our AWS scenarios. Our evaluation benchmark and all relevant code and instructions are provided in our aforementioned GitHub repository.

## C.10   Data format

The CSV file has the following columns: "Resource", "Prompt", "Intent", "Rego intent", "Difficulty", and "Reference output". An explanation of their intended use is provided in Table 6. NL refers to natural language, while Rego [35] and HCL [50] are expressive declarative languages.

Table 6: IaC-Eval dataset columns

| Column name | Descriptor | Format |
|---|---|---|
| Resource | All resources that this scenario requires. | NL |
| Prompt | User question to be fed to the LLM under evaluation. | NL |
| Rego intent | Validates the generated program for user intent fulfilment. | Rego |
| Difficulty | Calculated difficulty level (Table. 4). | Integer |
| Intent | Natural language intent for easy viewing. | NL |
| Reference output | Example correct program. | HCL |