# OpenReview forum: "IaC-Eval: A Code Generation Benchmark for Cloud Infrastructure-as-Code Programs"
_NeurIPS.cc/2024/Datasets_and_Benchmarks_Track — NeurIPS 2024 Track Datasets and Benchmarks Poster_

### Official Review · Reviewer_QEzy · 2024-07-25
**Interesting foray into IaC generation with Code LMs**

**Rating:** 7
**Confidence:** 4
**Correctness:** Yes
**Clarity:** Yes

**Review:**

In terms of originality and significance this work scores well. It sets out to solve a very hard problem (open domain infra code generation) and sets the first step for evals in an area where they are very hard to craft. In time, a task like IaC code generation could replace current hard benchmarks like SWE-Bench as this is an even more underspecified and open-domain task.

Overall, the weakness of the benchmark is its lack of true exec verifiability. The "tests" in IaC Eval are a bunch of handcrafted rules w.r.t existence of the required resources. The authors might have perhaps focused on creating scenarios where the existing resources could be tested by running the resultant service e.g. having an IaC intent for a database, where the tests are a bunch of insert queries and a schema followed by a select query returning the right set of records.

**Strengths:**

1. The authors set out to solve evaluations in a tough frontier for code generation model
2. The authors do show how current SOTA models can be lacking in code generation scenarios that span beyond simple completion that require reasoning at multiple levels.IaC generation is a hard task that needs understanding of networks, resources, security groups, and how they need to be wired to work together. In the future, a truly execution based eval in such a setting will be a gret test for agent-based code models.

**Additional Feedback:**

N/A

**Documentation:**

Yes

**Limitations:**

Yes

**Opportunities For Improvement:**

The biggest opportunity for improvement is to focus on deployable artifacts that can be functionally tested, e.g. requiring the deployment of a lambda with a provided function and then invoking the cloud function with some input, etc

**Relation To Prior Work:**

Yes

**Summary And Contributions:**

The authors investigate how good Code LMs are at generating IaC code and find a large scope for improvement, with the best models only scoring marginally below 20%. They also find that multi-turn and CoT style prompting improves their performance on open domain IaC code generation.

As a part of this effort, the authors release an evaluation set containing user intents as well as Rego "tests" that can be used to test the IaC code for whether it would enact the required resource changes.

---

> ### Author Rebuttal · Authors · 2024-08-17
>
> Thank you for your valuable feedback and support! We appreciate your suggestion. We would also like to clarify that our intent-specs extend beyond checking for the existence of resources. We also ensure that attributes are correctly configured and that dependencies and interconnections between resources are properly specified, all in alignment with the natural language intent of the problem, as detailed in Section 4.

---

### Official Review · Reviewer_VvUx · 2024-07-25
**Review of IaC-Eval**

**Rating:** 8
**Confidence:** 5
**Clarity:** Yes.

**Review:**

First, I'd greatly appreciate authors' effort in making one of the first IaC eval sets available. IaC is a critical, impactful, and high-demand skill, yet it is often overlooked by the community presumably due to two main reasons 1/ the difficulty in creating comprehensive IaC dataset, and 2/ the distint group between ML researchers and DevOps practitioners. Building a reliable IaC eval set would advance the field a lot, and promote better automation in infrastructure development.

Second, this paper is very well written, covering a lot of details with reasonable elaborations on many design choices. The dataset is in good size and the evaluation is comprehensive both in terms of metrics (e.g., pass@k vs LLM as a judge) and models used.

While I believe this is a great work that should be accepted, there are a few directions that the authors may consider adding or elaborating on:

1. On the evaluation metrics: authors mentioned that "...matched against the dependency graph using OPA [48], the most widely used IaC policy engine; a tool typically used to define policies (e.g., security, conformance), that IaC-Eval repurposes to evaluate user intent fulfilment." While OPA is a good tool, would performing a dry run of the generated IaC code on AWS be a more reliable metric?

2. On human expert: writing the correct IaC requires in-depth knowledge on cloud infrastructure. Could authors discuss how you obtained a group of human experts that can perform reliable annotation to IaC? In addition, how did the authors evaluate the quality of the annotations?

3. On security evaluation: have the authors considered adding evaluations for security risks in IaC? Given the importance of security in infrastructure management, this could be a valuable addition to the benchmark.

4. On language coverage: have the authors considered adding CDK/CloudFormation as part of IaC-Eval?

5. On prompts: could the authors discuss more how you obtain the prompts?

**Strengths:**

See Review

**Additional Feedback:**

N/A

**Correctness:**

There are a few avenues that can improve the dataset, but the current approach is sound enough as a first step towards IaC evaluation.

**Documentation:**

Yes, authors open-sourced the dataset and repo already.

**Limitations:**

See Review.

**Opportunities For Improvement:**

See Review

**Relation To Prior Work:**

The most relevant work (Xu et al. 2023) was discussed, however, I'd suggest authors discuss more on IaC itself as it is not very well known by the community.

**Summary And Contributions:**

This paper presents IaC-Eval, the most advanced IaC eval dataset to date. It includes 458 human-curated scenarios covering a wide range of popular AWS services, at varying difficulty levels, all in Terraform. Each example in IaC-Eval contains a natural language IaC problem description and an infrastructure intent specification. Results show that contemporary LLMs perform poorly on IaC-Eval, highlighting a need for advancements in this domain. This work addresses the growing need for reliable and efficient tools to automate IaC, which is critical yet often overlooked direction for modern cloud infrastructure management and DevOps practices.

---

> ### Author Rebuttal · Authors · 2024-08-17
>
> Thank you for your valuable support and feedback!
>
> > While OPA is a good tool, would performing a dry run of the generated IaC code on AWS be a more reliable metric?
>
> The benchmark actually does include a dry-run, which is conducted using the canonical “terraform plan” command, that ensures that the program has compiled successfully. This is performed prior to the OPA policy check.
> Performing full-scale deployments contradicts our benchmark’s goal of enabling scalable evaluation, as it can be time-intensive. Additionally, full deployments would not replace the need for the intent-based checks conducted by OPA, which ensure that the programs fulfill the natural language intent of the questions.
> We hope this answers your question!
>
> > Could authors discuss how you obtained a group of human experts that can perform reliable annotation to IaC? In addition, how did the authors evaluate the quality of the annotations?
>
> > Could the authors discuss more how you obtain the prompts?
>
> Thanks for the suggestion! We will include more details on this within our supplementary material (in addition to what we already have in Section C3) or the main paper (if space allows).
> Our co-authors learned from official AWS and IaC documentation, and other learning materials (including a mixture of public IaC repositories within GitHub, and IaC-related StackOverflow/Reddit posts). All configurations were deployed and tested to ensure correctness.
> Additionally, some of our co-authors are experts who have been using IaC for years.
> To maintain high quality, we have implemented channels (GitHub/HuggingFace issue trackers) for the community to report any issues/errors, which we will promptly address—this is explicitly mentioned in our supplementary material and repositories. We welcome contributions from the community.
>
> > Have the authors considered adding evaluations for security risks in IaC?
>
> This is a great suggestion and would be a valuable direction for future work. We have briefly mentioned this in our future work section (Section 5) and will provide more elaboration if the paper is accepted. For instance, it remains unclear how well LLMs perform in generating secure IaC code, automatically fixing security violations, or how existing security-based IaC checkers such as Checkov could be integrated into this process.
>
> > Have the authors considered adding CDK/CloudFormation as part of IaC-Eval?
>
> Good point! We currently focus on Terraform as it is the most popular IaC tool, but we will include integration of AWS CDK and CloudFormation as part of our future work, in the main paper as well!
>
> > The most relevant work (Xu et al. 2023) was discussed, however, I'd suggest authors discuss more on IaC itself as it is not very well known by the community.
>
> Thank you for the suggestion! Our paper is the first to explore IaC code generation with LLMs, thus we do recognize the importance of providing more context on IaC. If the paper is accepted, we will use the additional content page to include a dedicated subsection on related work focused on IaC, covering relevant research [7,8], industry tools such as security checkers [2,3,4] and other tools [5], as well as IaC frameworks [6].
>
> [2] KICS by Checkmarx. https://github.com/Checkmarx/kics/
>
> [3] Terrascan. https://runterrascan.io/
>
> [4] TFSec. https://github.com/aquasecurity/tfsec
>
> [5] Infrastructure Drift Detection. https://snyk.io/blog/tools-infrastructure-drift-detection/
>
> [6] SpaceLift. https://spacelift.io/
>
> [7] Julien Lepiller, et al. Analyzing infrastructure as code to prevent intra-update sniping vulnerabilities. In TACAS 2021
>
> [8] Qiu, et al. Simplifying Cloud Management with Cloudless Computing. In HotNets 2023

---

> > ### Comment · Reviewer_VvUx · 2024-08-24
> >
> > Thanks for the response from the authors. I don't have further questions.
> >
> > Good work deserves recognition, especially when the benchmark is well-considered, first of its kind, and bridges a critical gap between researchers and practitioners that have been largely overlooked. In light of this valuable contribution, I have raised the score to 8.

---

### Official Review · Reviewer_5a8w · 2024-07-30
**Solid benchmark work on Infrastructure Code generation with LLMs**

**Rating:** 6
**Confidence:** 2
**Correctness:** The proposed evaluation is well-thoug…
**Clarity:** The paper is well-written and easy to…

**Review:**

This is an interesting and well-motivated work as LLMs are becoming more popular in code generation and IaC is an important domain for developers. The dataset is well-constructed and the evaluation framework is thoughtful, considering different syntax/ambiguity semantics by compiling and checking against multiple possible infrastructure intent specifications. The paper also does a great job analyzing the dataset and provides a thorough evaluation including few-shot and RAG across closed and open-source models.

**Strengths:**

* The proposed evaluation framework is well thought out for the IaC code generation task, and the dataset is well-constructed from real-world scenarios in their usage of AWS services.
* Comprehensive evaluation including zero-shot, few-shot, and RAG across closed and open-source models.
* Good analysis of problems and results, showing the difficulty of the benchmark and the limitations of current LLMs in IaC code generation, including an error analysis.

**Additional Feedback:**

* How would you address the problem of library updates, e.g., new AWS API versions which may not be backward compatible with the current benchmark dataset? Would this make the benchmark harder to maintain and compare the performance of the models over time, e.g., the new model might use the new API and perform worse than the old model since the new API is not in the benchmark dataset?

* Could you provide more detail on the collection process of the dataset? How would it relate to real-world DevOps scenarios in the industry?

* Is it possible to translate the benchmark to support other cloud providers?

**Documentation:**

The data collection details are in the supplementary material and the dataset is open-sourced. I would actually suggest including more detail on the data collection process and the annotation process in the main paper. It's not very detailed how the data is collected and how it relates to real-world scenarios or just academic usage in their school.

**Ethics:**

No particular ethics concern that I'm aware of.

**Limitations:**

They discuss the limitation on only AWS.

**Opportunities For Improvement:**

* More examples and a systematic breakdown of the errors made by the models would be helpful to understand the challenges of this benchmark. For example, does the model struggle with syntax, semantics, or the ambiguity of the natural language description?
* The benchmark is specific to AWS services; it would be better if the benchmark could include not just a single commercial cloud provider but also other cloud providers or open-source IaC tools like Terraform.

**Relation To Prior Work:**

Yes, the paper discusses related work on code generation and coding benchmarks.

**Summary And Contributions:**

IaC-Eval is a new benchmark designed to assess large language models' (LLMs) ability to generate Infrastructure-as-Code (IaC). The dataset comprises 458 scenarios covering various AWS services, each including a natural language problem description and an infrastructure intent specification. The benchmark is designed to evaluate the LLMs' ability to generate IaC code from natural language descriptions. The output correctness is evaluated by checking against the ground truth infrastructure intent specifications.

Evaluation results show that most current LLMs, including GPT-4, perform poorly on IaC-Eval compared to standard code generation benchmarks where the score is mostly saturated. They have open-sourced the dataset and evaluation framework to facilitate further research in LLM-based IaC code generation.

---

> ### Author Rebuttal · Authors · 2024-08-17
>
> Thank you for your valuable support and feedback!
>
> > More examples and a systematic breakdown of the errors made by the models would be helpful to understand the challenges of this benchmark. For example, does the model struggle with syntax, semantics, or the ambiguity of the natural language description?
>
> Thank you for your suggestion! If the paper were to be accepted, we will include more concrete examples in addition to our example in Figure 6, either in the main paper (space permitting) or in the appendix.
> Currently, we do already include a systematic breakdown in Figure 5 of our main paper, which describes how existing models struggle with syntax and semantics through our compilation checks (shown in blue bars in Figure 5a), while evaluating the model's ability to understand the intent of the natural language description—serving as a proxy for ambiguity—through our intent specifications (shown in red bars in Figure 5a).
>
> > I would actually suggest including more detail on the data collection process and the annotation process in the main paper. It's not very detailed how the data is collected and how it relates to real-world scenarios or just academic usage in their school.
>
> Thank you for the suggestion! If the paper is accepted, we will have an additional content page in the camera-ready version where we can expand on this. In particular, we focus on use-cases that are mainstream, basic, and popular within AWS (as shown in Figure 2).
> Our data collection process was guided by (1) first learning from a mixture of public IaC repositories within GitHub, official AWS and IaC documentation, and IaC-related StackOverflow/Reddit posts, (2) crafting the required configurations, (3) deploying them onto the cloud, and finally (4) manually cross-validating the outputs to ensure correctness.
> We also do briefly mention our data collection process in Section C.3.1 within our submitted supplementary material, but will move them to the main paper as well.
>
> > How would you address the problem of library updates, e.g., new AWS API versions which may not be backward compatible with the current benchmark dataset?
>
> Good point! IaC is designed to abstract away low-level details, removing the need for users to directly interact with the underlying AWS APIs. While backwards compatibility is not exclusive to IaC and is a common concern in other programming languages, IaC does in fact maintain backwards compatibility, ensuring that older versions remain functional and usable.
> Further, breaking changes in IaC are rare, as such changes would impact all organizations using AWS, requiring them to rewrite their code.
> Typically, changes are minimal or are additive (new services, or additions to existing services). In the unlikely event that breaking changes do occur, we will augment our dataset to reflect the latest versions too.

---

### Official Review · Reviewer_nLxB · 2024-08-07

**Rating:** 3
**Confidence:** 3
**Correctness:** The claims made in the submission are…
**Clarity:** The paper is dense and difficult to f…

**Review:**

The quality of the paper is subpar, with several significant issues undermining its contributions. The clarity is compromised by dense and convoluted explanations, making it difficult to follow the methodology and results. The originality is limited as it relies heavily on existing benchmarks like HumanEval and EvalPlus, offering little innovation beyond adapting these concepts to IaC. The significance of the work is questionable, given the narrow focus on AWS services and the limited applicability of the results to broader cloud environments.

**Strengths:**

Novelty: The introduction of a benchmark specifically for IaC code generation is a unique contribution.

Comprehensive Dataset: The dataset includes a wide range of AWS services and scenarios, providing a robust basis for evaluation.

Open Source: The dataset and evaluation framework are made publicly available, promoting transparency and future research.

**Additional Feedback:**

The authors should focus on improving the clarity and organization of the paper. More detailed explanations of the methodology and results, along with a broader scope that includes multiple cloud providers, would significantly enhance the work's impact and relevance.

**Documentation:**

Yes

**Limitations:**

The authors have not adequately addressed the limitations and potential negative societal impacts of their work. While they mention the narrow focus on AWS, they fail to discuss the broader implications of their findings or propose ways to generalize their benchmark to other cloud environments.

**Opportunities For Improvement:**

1. The paper suffers from poor organization and dense prose. The methodology, particularly the intent specification and evaluation process, needs clearer and more concise explanations.

2. The work heavily borrows concepts from existing benchmarks without significant innovation. More effort is needed to distinguish this work from prior art.

3. The benchmark is limited to AWS services, which restricts its applicability. Including other cloud providers like Azure and Google Cloud would enhance its relevance.

4. The results indicate poor performance of current models but do not offer substantial insights or solutions for improvement. More detailed analysis and recommendations are necessary.

**Relation To Prior Work:**

The discussion of how this work differs from previous contributions is inadequate.

**Summary And Contributions:**

In this paper, the authors propose aC-Eval to evaluate the performance of large language models (LLMs) in generating Infrastructure-as-Code (IaC) programs. The benchmark comprises 458 scenarios covering a range of AWS services, and it utilizes intent specifications to validate the correctness of the generated code without requiring actual deployment. The authors claim that contemporary LLMs, including GPT-4, perform poorly on this benchmark, indicating the need for advancements in this area.

---

> ### Author Rebuttal · Authors · 2024-08-17
>
> > The paper is dense and difficult to follow.
>
> Thank you for your feedback! This is the first paper on IaC code generation, and thus we do understand that there would be quite a bit of prior background needed to fully comprehend the paper. If the paper is accepted, we will have an additional content page in the camera-ready version where we will clarify and expand on our explanations regarding IaC, the intent specification, and more background on existing works used in our evaluation (e.g., definition of pass@k). Please let us know if you have more detailed comments, we would be happy to address them during the discussion phase and make changes in the paper accordingly!
>
> > The originality is limited as it relies heavily on existing benchmarks like HumanEval and EvalPlus, offering little innovation beyond adapting these concepts to IaC.
>
> IaC-Eval is the first dataset and benchmark capable of evaluating IaC code generation. The only shared element between IaC-Eval and the aforementioned benchmarks is the use of the pass@k evaluation metric, a standard in code generation tasks.
> Designing our IaC-specific benchmark was a non-trivial undertaking.
> For one, developing our dataset required deep knowledge of AWS and IaC-specific constructs, including the intricacies of various AWS services and how to compose them into functional infrastructures. This process demanded over 1,720 hours of human effort.
> Moreover, our methodology for constructing the IaC-Eval was entirely different, incorporating compile-time checks and a unique intent-based verification tailored to IaC. We leveraged tools native to IaC, offering a practical solution without reinventing the wheel.
> Additionally, we conducted a comprehensive evaluation, by benchmarking against multiple state-of-the-art methods (as shown in Table 1 and Figure 4b), identifying where existing LLMs underperformed in IaC-Eval (Figure 5), and assessing the effectiveness of common enhancement strategies used in LLM code generation tasks (Table 2).
> This enabled us to produce the first comprehensive evaluation of how well current LLMs can generate IaC code.
> For these reasons, we believe therefore that IaC-Eval is a great fit for the NeurIPS Datasets and Benchmarks track.
>
> > The benchmark is limited to AWS services, which restricts its applicability. Including other cloud providers like Azure and Google Cloud would enhance its relevance.
>
> The core concepts of our benchmark (e.g., intent specifications), are applicable across other cloud platforms such as Azure and Google Cloud because IaC is built to be cloud-provider agnostic.
> As this is the first benchmark and dataset focused on IaC, we chose AWS as it is the most popular cloud provider, and is broadly adopted by the community.
> Our dataset focuses on AWS services because creating these datasets is time-intensive and requires an in-depth understanding of each cloud service in concern, especially since we cover a wide range of services and include questions of varying difficulty levels.
> We have already noted the inclusion of other cloud providers as part of our future work, and will expand on it with the reasons aforementioned.
>
> > The authors have not adequately addressed the limitations and potential negative societal impacts of their work.
>
> We will make sure to include a section discussing the social impacts of our work. In that section, we will emphasize that we do not foresee any negative societal impacts arising from our research:
> while IaC-Eval does evaluate LLM-generated IaC programs, it’s important to note that these evaluations are limited to compile-time operations and do not involve deploying the programs into the cloud. This approach ensures that users need not worry about any potential impact on their cloud infrastructure. Additionally, we have conducted thorough manual reviews of IaC-Eval’s dataset to confirm that it is safe for execution, posing no risk to either cloud providers or users.
>
> Moreover, the composition, collection process, and other relevant details of our dataset are provided in our supplementary material, which adheres to the “datasheets for datasets” format to ensure transparency and accountability.
>
> Finally, we have included the limitations of our work in Section 5 of the main paper and will expand on these points with further elaboration in the final version if the paper is accepted.

---

### Author Response · Authors · 2024-08-29
**Thank you for the responses and looking forward to discussions**

We sincerely thank Reviewer VvUx for their responses to our rebuttal and look forward to further discussions with Reviewers nLxB, 5a8w, and QEzy to address any potential remaining concerns over the next 2 days. Thank you in advance!

---

### Decision · Program_Chairs · 2024-09-26

**Decision:**

Accept (Poster)

**Comment:**

This is a benchmark for Terraform, a high-level framework (config programming language) for cloud infrastructure provisioning. This is a very challenging problem and the authors did a good job explaining the problem and creating a challenging benchmark.

One limitation is that the problem is somewhat narrow, but still its very useful for infra config generation. It is also quite interesting that current frontier models do so poorly, indicating a lot of room for improvement.

I think the lack of real executions for the test cases is a limitation but I think the significant cost they would require justifies it.
Overall I think this is a very interesting paper in a narrow but important problem. The benchmark is not ideal but still has good insights and is an important first step.